# Use and Design of Chatbots for the Circular Economy [note 1]

**DOI:** 10.3390/s23114990

**Published:** 2023-05-23

**Authors:** Răzvan Daniel Zota, Ionuț Alexandru Cîmpeanu, Denis Alexandru Dragomir

**Affiliations:** Department of Economic Informatics and Cybernetics, Bucharest University of Economic Studies, 010552 Bucharest, Romania; zota@ase.ro (R.D.Z.); denis.alexandru97@gmail.com (D.A.D.)

**Keywords:** artificial intelligence, chatbots, circular economy, recycling, human computer interaction

## Abstract

The fact that advanced technologies and their economic applications have generated increasing resource costs justifies the transition from a linear approach to a circular one in order to control these costs. From this perspective, this study presents how artificial intelligence can help achieve this goal. Therefore, at the beginning of this article, we begin with an introduction and brief review of the literature on the subject. Our research procedure involved the combination of qualitative and quantitative forms of research using mixed methods. In this study, we presented and analyzed five chatbot solutions used in the field of the circular economy. The analysis of these five chatbots helped us design, in the second part of this paper, the procedures for data collection, training, development, and testing of a chatbot using various natural language processing (NLP) and deep processing (DP) techniques. Additionally, we include discussions and some conclusions regarding all aspects of the subject to see how they can help us in future studies. Furthermore, our future research with this topic will have as the goal the effective construction of a chatbot dedicated to the circular economy.

## 1. Introduction

In recent years, there have been many developments in the field of the circular economy (CE). Additionally, a series of models that merit investigation and analysis have been developed for CE. CE models involve activities such as reduction, reuse, repair, rethink, redo, recovery, and redesign, as well as related approaches. Moreover, CE may benefit from research in the fields of economics, chemistry, biotechnology, and Information Technology and Communications (ITC). In this context, this article presents an analysis of how ITC can help the consumer position transition to a CE approach, especially when it comes to recycling materials and equipment.

In the last two centuries, human society has gone through an industrial revolution. This has brought about a never-before-seen level of wealth, as well as commensurate benefits in the well-being of people in countries undergoing this process. The resulting system, however, is significantly more complex than anything in history and thus much more difficult to understand.

Today, however, if we are to maintain our ambition of development and scalability, we must take into account an exhaustive understanding of the system under which such objectives can be achieved. The circular processes of development, maintenance, design, and feedback that are present in such a structure are constantly expedited by emerging new technologies that play a crucial role in reinventing the important general aspects of such an economy.

By implementing circular economy solutions, it is hoped to improve sustainability by reducing the consumption of natural resources and the energy required for production, minimizing the amount of waste and polluting emissions, as well as increasing economic efficiency and long-term competitiveness. In addition, CE can provide significant benefits in terms of economic growth and innovation, by stimulating the development of new technologies and sustainable business models, as well as by creating new employment and market opportunities. Therefore, the circular economy will contribute to the building a more equitable, sustainable, and prosperous economic system for all.

Consequently, new functionalities and adjuvants have been developed to ensure that the only impediments faced in achieving benefits from CE are either material (the finite resources that we must manage and work with) or pertaining to vision (we have all that we need to build, but what exactly are we doing?). One perfect example is represented by artificial intelligence (AI), which can offer help in switching from a linear model and way of thinking to a circular one, beneficial for the environment. This also implies the move from a fixed perspective to one that is more flexible.

This raises the question: “Is there a way in which AI can help the transition to a circular approach?” In this regard, several studies have been conducted that show the importance of AI, and the conclusion is that AI can deal with complex problems and analyze a large amount of information efficiently. To put it bluntly, three main methods were discovered to enable CE and sustain innovation:Circular development—from products and modules to materials and components. This is achieved by iterative learning processes performed by machines;Optimization of the circular infrastructure—required by the materials that will enter and be reused throughout the system;Predictive management and maintenance—used in the work with circular models in business.

In a highly digitalized society, there is increasing interest in the use of AI methods, techniques, and algorithms [1]. Furthermore, from all ITC subdomains, we have focused on how AI can lead to progress in the case of CE, using chatbots [2].

CE is an area where smart applications are still in their infancy. After searching Google Scholar, we noticed that there are few articles on chatbots implemented in the CE field where intelligent solutions are described in detail. In addition, the solutions found presented few truly useful options for the protection of the environment and nature. However, the five chatbot applications that we have identified provide sufficient information regarding the improvement of citizens’ activity regarding the contribution to the circular economy. We note that, for the description of the intelligent solutions found, we searched for information from several articles, websites, and publications; thus, we expanded our research field and managed to collect information from various reliable sources.

In conclusion, starting with the definition that states that ‘chatbots are online computer systems that can mimic humans and converse with humans using natural language’ [2], we analyzed five interesting chatbot applications that can help the evolution of CE. First, we have presented the characteristics of AIRe, a chatbot developed by TheCircularLab (Spain). Secondly, we analyze a chatbot used in the city of Yokohama, Japan, to clarify issues related to the selective collection of waste by people. The third example comes from France, analyzing Trizzy, with its main objective to help people better understand the concept of waste management (sort it, categorize diverse types of containers, and how manage e-waste, such as exhausted batteries). The last two examples are BioHiTech (a chatbot application used to reduce food waste) and Eco-bot, a chatbot that acts as a virtual energy-saving assistant.

Furthermore, we developed our own way of designing a chatbot solution that can be used in the recycling field, and, in the end, we mentioned some conclusions and probable future developments of our research. Furthermore, the present article represents a follow-up to the investigation carried out in the research paper titled ‘Using artificial intelligence for the benefit of the circular economy’ [3], which explored the various ways in which AI can facilitate the attainment of circularity within an economic framework.

## 2. Materials and Methods

If we consider the existing literature, we can define a circular economy in a specific and understandable way. The circular economy represents a system or structure that has the objective of imposing four main laws on resource management and circulation across different cycles. The circular economy has two main principles on which it is based: attempting to enhance the reusability and efficiency of its inherent elements and materials and putting an end to the lack of usage regarding the former. We can group these four ideas into ‘the 4 Rs’: remanufacturing, repairing, recycling, and reusing.

The concept of CE has been defined by multiple authors and appears in a multitude of scientific papers. For example, F. Sariatli, characterizes this concept in his work entitled “Linear Economy Versus Circular Economy” [4] as: “An economic system focusing mainly on the conjuncture in which there are flows and streams of material that must have low levels of contamination and must be circulating with a proportionately tremendous velocity, entering and interacting with sensitive areas (the biosphere, for example) only if they are described by nutrients”. In other words, CE is an industrial economy by nature and is regenerative and pure in its intentions [5].

Other authors describe this concept; an example is represented by W.R. Stahel, who, in his work entitled “The circular economy” [6], mentions that there are four main components on which it is based:Networks, final users, and companies must be very well connected, and there must be effective communication between them;Companies in both the private and public sectors must adopt new circular business models to generate value;Products are the level of action situated at the bottom, the most granular part that specifies how actors must be designed: clean and in such a way that they can be reused. In other words, according to your environment;Circular growth must be enforced and maintained by market regulations and incentives.

To facilitate the evolution from a linear economic model (the predecessor of the circular) to a circular one, a sequence of changes and measures must be considered. Changes that do not produce beneficial effects are of no use. It would be a waste of material, energy and time. Instead, such a system must be resilient over time and offer numerous opportunities. In this sense, a tool that we can use is AI. This, together with the improvement of business processes through digitization, can be used as one of the crucial factors for the benefit of the circular economy [7].

Here are some examples of ways that AI can be implemented in organizations and companies to empower circular standards:Clustering, the process of creating groups using various and multiple data points obtained from a single ‘source of truth’, whether we speak of data or information;Time-series analysis, the process of characterizing a variable evolution across a determined period. To exemplify this, imagine monitoring the evolution of a specific resource across a passing cycle in the circular economy or the resource consumption over a specified period (one year, two years, etc.);Object detection involves the process of discovering whether a specified object can be found in an image or not. As an example, we can use this mechanism to find the materials from which a food container is made of, as well as to classify them and store them in their respective collection and treatment points;Chatbots (the subject of this article) can exercise human-specific behaviors, such as basic communication, answering questions, addressing new ones, providing information, and continuously learning using complex algorithms;Natural language understanding can be used to determine concepts based on from natural language produced by humans [8].

A chatbot is an AI solution designed to converse with humans. This conversation can be verbal, as in the case of voice queries, or based on written text [9]. Chatbots are valuable tools for information acquisition and can be accessed through a variety of devices, including computers, laptops, tablets, mobile phones, and smartwatches, all of which require Internet access. The chatbot has the ability to provide answers to user queries and can initiate discussions based on question and answer patterns that it has incorporated into its knowledge base. Basically, it is an IT solution that incorporates AI and uses natural language processing (NLP) [10].

In addition to user natural language input, a chatbot contains the knowledge base, which provides it with a source of truth. The chatbot’s knowledge base allows it to find answers to user queries by using specific methods to search for requested information, such as analyzing the similarity of words, comparing them to the basic root of each word, derivation words, their composition, structuring, and classification of words in an utterance into parts of speech or parts of a sentence [11]. Furthermore, the chatbot is able to understand the meaning of words in utterances and generate appropriate responses, even if the user’s phrasing is incorrect or some words are not typed correctly, by comparing the query with other similar questions and providing a response that is close to what was understood.

When words describing emotional experiences, emotions, and feelings are used in recorded messages, the chatbot uses emoticons, drawings, icons, or other representations of these emotional experiences to communicate an answer to the user’s query [12]. Through programs and algorithms, the chatbot mimics human intelligence and is able to understand the user’s needs. The chatbot is capable of supporting multiple conversations at the same time and providing each user with answers, even if they are from different fields of activity. The chatbot analyzes the user’s intent based on the words used in the text, the user’s access history on the app, the features the user uses and what they want to achieve. Specific algorithms are used to synchronize different applications used by the user, and when a change is made to a single application that is related to several applications, the chatbot adjusts accordingly [13].

Chatbots have the potential to serve as a promotional instrument for entrepreneurs and companies by articulating the nature of their services, the benefits and features of their offered solutions, the method of their execution, the advances implemented and the mode of application of these solutions in various fields of operation. Chat dialogue can also be used to promote product services and how to access them of these services [14]. Generally, developers design the IT solution to meet the needs of the product users, paying attention to the needs of the company, to the number of people who would use the IT solution, the increased benefits of this solution, and the price of the IT solution related to the demand and supply of the labor market [15]. Being a robot, the chatbot is based on a program and the constant ability to learn both from the dialogues it has with users and from the database and information sets at its disposal. However, the logic that the chatbot has and the way it works are different from human logic. Therefore, the way the intelligent robot learns allows it to be constantly improved.

In our research activity, we chose to use methods based on qualitative research and mixed methods. We analyzed chatbot examples implemented in the circular economy in both Europe and other countries of the world, especially focusing on originality, creativity, and the scope of the solution. These chatbot examples focus on the transformation/modification of the initial IT solution by other developers so that the benefits are multiple. Moreover, chatbot users can be as many as possible, such that the result of the implementation can bring benefits in sectors of activity adjacent to the implementation sector of the chatbot.

Among the methods we used in our qualitative research, we note the following:-Case study;-Participant observation;-Thematic study of documents [16].

We focused our research on countries that have integrated AI solutions in their circular economy and examined the various ways in which these solutions were implemented, as well as the impact they had on their respective work environments. Furthermore, our research involved conducting online searches, watching videos that showcased online platforms used in the circular economy, and analyzing articles on online platforms in this sector. We placed special emphasis on online platforms that had integrated a chatbot, which assists users in understanding the importance of recycling and reusing various materials that would otherwise be discarded as waste.

Moreover, we investigated the chatbot’s ability to aid in collecting various waste materials at designated points established by the authorities, sort the waste by category, identify where this waste could be brought, and provide additional information on the proper handling and safe disposal of hazardous waste in specially designated areas.

The systematic analysis involved a thorough examination of specialized articles, videos, and chatbot courses using various programming languages. The documents were meticulously studied and their contents were carefully sorted to ensure that only relevant materials were retained. These materials were then used to inform our thinking, as we sought to build and develop an IT solution that would meet our desired objectives, with the ultimate goal of successfully applying it in practice.

The research procedure also involved combining qualitative and quantitative forms of research using mixed methods. The mixed-method strategies we applied were sequential mixed-method procedures. These procedures are based on an explanatory sequential design and are performed in several steps. These steps are as follows:Tracking, finding, collecting and analyzing quantitative data on online platforms used in the circular economy;Tracking, finding, collecting, and analyzing qualitative data on online platforms used in the circular economy. This step is based on the results obtained after sorting the quantitative data; that is, step 1 focuses on the group;Identification of weaknesses: a lot of time to obtain information, analyze, compare, and evaluate data;Identification of strengths: simple, original, and creative design, the ability to transform/add additional options to the implemented chatbot so that it can perform more actions and be used in other sectors of activity.

All references used in this paper were selected from prestigious journals, which contain recent studies and qualitative analyzes that have concrete information about the process of modeling, creating, and implementing a chatbot, as well as its relationship with a circular economic environment (how can it be integrated into the latter and the value that it creates).

## 3. Results

One of the many applications of AI is the chatbot, which is designed to interact and communicate with humans in an automated manner. Depending on their implementation, these chatbots can provide users with useful information about a specific topic and possess various functionalities that are intended to enhance and enrich the user experience. For example, the ‘entity recognition’ pattern can be used to identify various objects and distinguish them, which can help to sort and recycle waste more effectively from a circular perspective.

The potential of chatbots in the future is significant. Instead of relying on websites or search engines to find the information they need, users can engage in human-like interactions with intelligent chatbots at every stage. This potential also applies to the circular economy, and to demonstrate this aspect and showcase what the market currently offers, we have chosen five chatbots to analyze. Each of these chatbots has common functionalities, as well as specific ones tailored to the circular subdomain in which they operate (whether it be energy saving, recycling, etc.). Without further ado, here are the five chatbots.

### 3.1. Analyzed Chatbots

#### 3.1.1. A.I.R-e

TheCircularLab, in collaboration with Accenture, has developed a chatbot that focuses on enhancing the circular economy, known as the Asistente Inteligente de Reciclaje, or A.I.R-e (the intelligent recycling assistant). This chatbot incorporates various recognition patterns, including text, voice, and image, which, in combination with its default behavior, aids users in the context of the circular economy, specifically in the areas of reusing, recycling, remanufacturing, and repairing. The A.I.R-e chatbot represents the culmination of efforts by numerous individuals aimed at increasing public awareness and educating the masses on the implementation of eco-design and efficient waste management within the current economic structure, while also endeavoring to promote entrepreneurial development. Using state-of-the-art technologies, this chatbot is able to identify the material composition of specific containers, thus simplifying the sorting and recycling processes for materials such as textiles and wood [17].

Upon examination of the official product description, it becomes evident that, while a growing number of people are recognizing the importance of implementing circularity through recycling and its associated benefits, many lack practical knowledge on how to translate this awareness into action. Certain products, for instance, may pose challenges in terms of identifying appropriate disposal methods because of the difficulty in discerning their constituent materials. In this context, the chatbot under discussion is well equipped to analyze and respond to such inquiries, thereby enhancing user understanding and promoting more effective circular practices. Let us give an example: If we ask ‘Where should I dispose of the coffee capsules?’, the answer will be something similar to the following: ‘Cafe recipients must go to the gray container or to the closest clean point. Speaking of which, within a few meters, I can see some collection centers. I can sort them by brand and popularity and give you the names and basic descriptions. Overall, you should go to their website and see which is the most viable for you. Are you satisfied with this response?’ [18].

As stated already, the bot, in addition to the base functionalities that make it a highly informative instrument (the capacity to learn over time from input data and from its mistakes, to emulate human interaction, to respond to different questions asked by the user through different data processing algorithms), also has image and voice recognition. If we are to give a brief definition of it, we can say that image recognition represents the process of identification using the visual aspects of specific things in a larger picture. For example, it can identify a football glove in a locker, also giving the basic description of it and the category to which it belongs. This alone is a door that can open many opportunities to reduce waste and introduce material into the economy through intelligent practices and processes. In addition, the tool is always open to learning through self-learning techniques and will become increasingly autonomous over time, making better decisions as it develops [19].

#### 3.1.2. Iio

IT specialists, people with experience in the field, enthusiastic, creative, entrepreneurial, and innovative, prove that the human mind has no limits and direct its efforts to obtain benefits in areas where exhausting resources could create problems soon and in the long term. They find solutions so that many materials already used can be reused and information can reach all citizens of a country quickly and easily. We are talking about the reuse of objects, the collection and the recycling of certain categories of materials that are currently stored in landfills and pollute the environment and the health of all living things.

One such smart solution implemented in the Japanese city of Yokohama is the Iio chatbot, an application used in CE through which all city residents, led by the mayor, collect, sort, and store used objects. These old objects can be old appliances, bottles, jars, plastic bins, paper waste, cloth, used textiles, small metal items, edible waste, used batteries, and other objects that may have been thrown into the city landfill. The idea was to find a solution that would help the economy by finding different resources/materials that could be used as raw materials, given that the depleted resources of the planet are ending. Without finding ways to replace these raw materials, the economy would stagnate, many people would lose their jobs, and many areas of activity would disappear. The benefits of recycling objects/materials that should have been disposed of are huge. On the other hand, without these innovative ideas, every country would become a landfill and the planet would die slowly. Japan understood this problem and channeled some of its funding to investigate the circular economy, informing the public about what it means to collect and recycle waste, what the benefits of these actions are for people, animals, and plants, and a cleaner environment [20].

The chatbot implemented in the city of Yokohama by the Office of Recycling and Environment has many smart options that can answer local questions. These questions are about recycling garbage, where and how to store recyclable items belonging to distinct categories, how to separate garbage, where large objects (that people no longer use) are brought in, which are the objects that cannot be burned, and where old appliances can be brought. When a person does not know how to sort trash correctly, they will query the Iio chatbot. All the chatbot user has to do is enter the name of the trash item in a written or verbal message to the chatbot and they will receive a reply about what to do, where to store it, day of the week, time and place in that garbage item must be stored. The chatbot also provides tips on dangerous items that are thrown in the trash, such as a lighter. If the chatbot notices that a person is throwing a lighter in the trash, he says, “Make sure there is no gas residue in the lighter because it can cause a fire at the collection site!” [21].

The recycling rules in the city of Yokohama require residents of the city to bring everything they no longer use and throw it at these fixed points every day of the week, before 8 am, in places specially designed for each category of waste.

In the chatbot database, the specialists included information on garbage in the city of Yokohama and Repl-AI, a chatbot technology developed by the telecommunications company NTT Docomo Japan. The chatbot’s knowledge base includes keywords from the circular economy, as well as dialogues between people who address these topics. This chatbot was originally designed to raise awareness of recycling among the population, but also to give people more knowledge about how various materials can be sorted and how they can be successfully recycled and reused in other areas.

The creativity of this solution is admirable, given that Repl-AI technology helps the chatbot answer correctly and amazingly wise questions that have nothing to do with garbage. In this case, all keywords in the database are used and the chatbot makes logical connections with all the words included in its program. One such example that was repeated and debated both in the press and on social networks, in articles with specific content, was where the word “husband” was used in a message written to a chatbot by a user. Instead of a 404 error that should have appeared in response to the chatbot not being able to return a message to the user, the chatbot sent a clever response that surprised everyone: “Armand Salacrou once said,” People get married because of lack of judgment, get divorced because of impatience, and remarry out of lack of memory. ‘Why not try to train your patience?’”” It is good to know that the chatbot would have sent the same response when the word ‘husband’ was used in the user’s message to the chatbot and when it found words such as ‘marriage’, ‘beloved’, ‘life’, and ‘anger’ in other written or verbal messages [22].

#### 3.1.3. Trizzy

In addition to the Spanish chatbot A.I.R-e, whose focus is on virtual recycling, there is another similar platform developed by the French company named Trizzy. Trizzy’s main objective is to help humans better understand the concept of waste management (how to sort it, how to categorize diverse types of containers, how to administrate electronic waste, such as exhausted batteries) and to avoid throwing the former away together with other junk belonging to the household. Trizzy also has a similar base functionality as any other platform in this area, meaning that it can search and provide useful information not only regarding the operation mentioned above, but also about nearby recycling centers. Through self-learning technology, it can adapt to user preferences and customize its supplied tips to achieve the definition of zero waste [23].

Trizzy is the first zero waste assistant that supports communities and businesses around the world in managing and minimizing waste, promoting circular economy, social bonds, and implications and reuse.

Waste—because good waste is no waste at all, changes in behavior can have a significant impact.Circular economy—because one response to implementing clean practices is the circular economy, reuse is promoted through the initiatives present in the regions.Social bonds—social bonds play a crucial role in facilitating interpersonal relationships and promoting stronger ties between citizens and their local ecosystem. These social bonds not only provide a sense of belonging and emotional support, but also facilitate access to information and resources within the community. Furthermore, strong social connections can help strengthen the interaction between citizens and their local ecosystem by promoting environmental awareness and encouraging sustainable practices. Therefore, the cultivation of social connections is an essential component of building strong and thriving communities.

To better visualize the capabilities of this software, let us take a little detour, talk about e-waste, and try to understand the problem it presents a little bit. To be precise, what is it? E-waste, also known as ‘electronic waste’ or ‘end-of-life electronics’, is the term used to describe the kind of waste consisting of electronics that have served their purpose and are at the end of the use cycle. Because of that, they are often given to the nearest recycler. As useful as they might seem to be, without adequate incentives and standards, improper or wrongly applied good practices can lead to health and environmental problems, even in developed regions, where facilities have the processing power and abstract most of this work exist [24].

However, the most affected are developing countries that have already manifested serious issues that cause harm to human health and the surrounding environment. If we are to give examples, we can advertise our attention to the fact that colleagues are exposed to harmful materials and substances due to the methods used to scrap electronic components (acid and open-air burning baths). The latter can create exposure to some highly damaging materials, such as arsenic, mercury, and lead, which can ‘lead’ to irremediable effects, including several types of cancer, neurological disorders, and decreased IQ. The rough estimate of the Environmental Protection Agency (EPA) states that in 2009, the total discarded electronics (computers, mobile phones, peripherals) in the US was approximately 2.37 million tons. However, the worrying fact is that only 25% of this amount was recycled, the rest being thrown into landfills, making it impossible to recover the metal present there [25].

Returning to the platform, as we have seen above, waste management is one of the main pillars of the process of keeping the environment clean and cannot be overlooked. CSR policies are increasingly established in every company, and there is a need for real implementation of strategies. According to the official website: 

“Each office worker produces an average of 130 kg of waste per year at his workplace. It is especially important to dispose of the latter and try to reduce it as much as possible. However, if the right thing is not always easier to do, even at home, it becomes even more complex in places such as workplaces because the required information is not always accessible to the final users. But no worries! All you have to do is ask Trizzy! Do you have to manage industrial waste and/or waste specific to your activity? We can create a custom waste database so that Trizzy only recognizes your waste. In addition, Trizzy can also suggest how to store the scrap and prepare it for shipment. To achieve your goals and meet your obligations, you need to obtain the correct information from a reliable source and at acceptable time intervals. We want to facilitate that for you and make the information easily accessible to all your employees.”[26]

Of the 2.4 million tons of waste produced only in France, only 35% of the paper waste goes through proper recycling. The room for improvement is consistent and, inevitably, better and correct information. One little downfall is the fact that the only language available at the moment is French. In addition, Trizzy, the startup, offers a chatbot that is designed to help all the corporate companies and social communities that it can. From the availability point of view, it is available on social networks, websites, and mobile applications. Trizzy is specialized in zero waste policies, collection schedules, sorting particular kinds of waste, and more. They also provide functionality in which you can place a QR code in your bins to launch the application and interact instantly [27].

#### 3.1.4. BioHiTech

The CE area is attracting increased attention because it is able to summarize innovative and efficient smart application implementations. One such model was launched in September 2016 in the city of New York in the USA by the company BioHiTech Global [28]. This company is specialized in providing technology solutions for waste management and sustainability. It is known that a large part of the food purchased from each family is thrown in the trash every week. Family members cannot consume the entire amount and food cooked in much larger amounts than the family needs is thrown away. This is about responsibility and managing the food needs for each family according to its consumption and the needs of each family member. If this is performed responsibly, each family would buy exactly what they need and waste would be eliminated.

The intelligent solution implemented by BioHiTech Global (BioHiTech Alto) [29] uses information and analysis to achieve sustainable business processes where decisions are responsible and intelligent. This intelligent solution can be applied to any industrial machine, or for any industrial equipment connected to the Internet. The implemented chatbot uses text messages to monitor, manage, and control industrial machines [30]. The platform is equipped with an on-board scale that has the role of weighing food waste, identifying its type, removing the category of waste that is not part of food waste, and adding other food waste.

The chatbot provides customers with information on collected food waste and efficient ways to use industrial machinery and equipment, being a tool for customers to manage and reduce food waste [31]. The impact on the large amount of food waste is significant; the chatbot also provides information about the storage areas for this waste, and what options can be considered for the biological treatment of food waste both in the spaces recommended for the storage of this waste and in areas identified by employees or customers who are off-site. This smart application can be said to be among the best in providing zero food waste solutions in businesses, companies, cities, highly populated areas and high staff turnover.

Families and restaurants, hotels, cafes, and other places where food consumption is carried out in large quantities manage to collect tens or hundreds of thousands of kilograms of food waste per year. As companies learn to use AI and incorporate it into current and future technologies, we can talk about efficient disposal of food waste. The chatbot acts as a proofreader that provides immediate access to all collected data and identifies any type of food waste, identifies inefficient behaviors that lead to high waste production, analyzes information and reduces the costs of processing this waste. Many hotels, restaurants, and cafes in the US have implemented this chatbot, managing to see the enormous potential in any type of industrial equipment.

The technology is connected to the food waste analysis platform. The company’s research also plans to connect with Cirrus’ mobile analytics app in the future. Customer communication with smart machines is performed via text, and for the future collaboration with the Slack messaging platform is desired. Among the modern technologies for processing food waste is mechanical biological treatment, a revolutionary technology that is also being implemented in a new facility that the company wants to build in West Virginia.

#### 3.1.5. Eco-Bot

The Eco-Bot package represents a chatbot that helps the user by being his virtual energy-saving assistant. The Eco-Bot manages to engage with the one who is using it in almost real-time communication, as it uses artificial intelligence to benefit from the constant feedback, enhancing its abilities. The chatbot is co-founded by the European Union through the Horizon 2020 Program [32].

As the main functionalities, the chatbot can provide accurate data on energy consumption (for a specific interval and appliances) and related costs. The chatbot also has a ‘target setting’ option to help make the distinction between different user groups and to recommend each of them personalized energy savings ideas. The chatbot serves both professional users (such as building managers) and private establishments. The chatbot can be used as a standalone app or integrated into the website of a utility.

When it comes to the structure, the application consists of a backend and a frontend. The former has the following components:An operational component that includes a rule engine—it enables the execution of different business rules.A behavioral model.NILM (non-intrusive load management) algorithms for residential and several types of non-residential buildings that give an estimate when it comes to electrical consumption of large commercial loads or individual appliances. As input, it uses only smart meter data, without resorting to data such as demographics. NILM comes with improved smart home automation, targeted energy efficiency feedback, improved demand response, and customer safety feedback.

The markets targeted by this platform are those that need of an energy market expertise chatbot. Through its behavioral model, the Eco-Bot project will classify already existing socioeconomic energy consumer models resulting from and derived from projects and research conducted in Europe and internationally and will produce a classification of these models, while mapping them to actual market segments. Chatbot engagement strategies will be constructed on an exhaustive analysis that includes social and cultural factors that are crucial to the formation of energy-saving behaviors among commercial and individual consumers. Moreover, the Eco-Bot project will pay special attention to soft behavioral aspects, the latter being especially important for pro-ecological perspectives among the final users [33].

Regarding energy consumption, the application will not use special sensors or sub metering, as it can be expensive, especially since the number of residential and commercial buildings is rapidly increasing. On the other hand, energy disaggregation provided via NILM brings to the table a non-intrusive, purely computational, software-based approach to separate aggregate consumption from a single electricity meter into individual appliance loads [34].

#### 3.1.6. Chatbots Comparison and Analysis

In order to promote the adoption of circular economy principles, commonly referred to as green principles, and to benefit both the environment and society, it is important to develop solutions that facilitate consumer understanding and application of these principles. Such solutions should provide clear and understandable information, be easy to use, provide a secure and confidential environment and be specialized in circularity.

This review aimed to examine current market offerings for circular economy solutions and assess their impact and capabilities. Through this investigation, we have identified five solutions and evaluated their strengths and weaknesses in meeting the criteria mentioned below.

First and foremost, security is critical in a chatbot application because it protects users’ privacy and prevents unauthorized access to their personal information. Chatbots can collect sensitive information from users, making it essential to securely store and protect this data from hacking and fraud. Ensuring the security of a chatbot helps maintain user trust, comply with regulations, and avoid legal liabilities. To achieve this, developers should implement best practices, such as encryption, access controls, user authentication, and regular security audits, and stay informed of the latest security threats and vulnerabilities.

Secondly, the way a chatbot interacts with users is crucial to its success, and dialogue is a key component of that interaction. Good dialogue can make the chatbot more engaging and user-friendly, leading to higher user satisfaction and a more positive brand image.

Additionally, the user interface (UI) of a chatbot is an essential component that impacts how users interact with and perceive the chatbot. A well-designed UI can improve user engagement, positively influence brand image, and create a better user experience.

Data accuracy is also very important, as it directly impacts the chatbot’s ability to provide accurate and relevant responses to user queries.

Speaking about *NLP* technology, all analyzed chatbots are using it.

The scores attained by the evaluated chatbots are tabulated below. These scores are derived from the collective experience of the users, as well as their corresponding reviews. Specifically, a score of 8 (out of 10) implies that 80% of the total reviews related to a specific indicator were positively rated.

As an example, A.I.R-e has generally received positive reviews for its ability to provide helpful information about recycling and waste management. Users have appreciated its user-friendly interface and conversational approach, making learning about recycling more accessible and engaging [35].

Chatbots functionalities can be seen in Table 1.

### 3.2. Chatbot Design

#### 3.2.1. Introduction

Based on the examples that we will analyze, we present in the following section the steps needed to develop a chatbot; therefore, the first step in this case is to establish the domain in which it will be implemented. This is followed by a long and sustained activity in which information is collected in the field in which we want to implement the chatbot. The quality of the information collected is analyzed, interpreted, classified and selected by analysts who can determine whether the data are sufficient and qualitative for the development of a chatbot or if additional data are needed to achieve the proposed objectives. The next step is to establish the techniques that will be used to build the chatbot.

‘Deep processing it is one of the extreme ends of the level of processing spectrum of mental recall through the analysis of the language used. To create a much stronger memory trace deep processing requires the use of semantic processing’ [36]. Developers use this AI technique to create a recurrent neural network based on which responses are generated.

In the following, we describe how a chatbot model can be built, using NLP, DP and the SeqtoSeq model of recurrent neural networks (RNNs) [37]. We establish different NLP techniques that will be used to process and prepare the data collected for chatbot training. These techniques will help the chatbot better understand the user’s intent. Once the data collected using NLP techniques are processed, the SeqtoSeq (chatbot brain) model will be implemented.

The Seq2Seq model is used in virtual assistant chatbot solutions [38]—it uses sequence-by-sequence modeling and has an encoder-decoder architecture built using single or bidirectional long short-term memory cells. Figure 1 shows the architecture of this model. SeqtoSeq loads the model sentence (encoder) for the first time to understand the user’s intention. For the best answer, each word is assigned a number. Thus, all the numbers resulting from the encoding process will form a vector. On the basis of this vector, all words in the selected sentences are classified, and this model classifies the importance of each word in the source sentence. In this way, the first word is obtained from the answer to be given to the user. This process is repeated until the entire sentence is ranked according to the importance of each word, and the definitive answer is sent to the user.

A chatbot is a program whose role is to answer user questions. This is a conversation that takes place between users and the robot, in which the latter automatically learns from all the dialogues that take place and constantly improves its knowledge base. We wonder ‘how the chatbot manages to talk to a lot of users by building messages for each question and making connections with all the information it has?’ [39]. Here is the representation of this process in Figure 2. In this diagram, you can see how the user asks different questions to the chatbot. The chatbot searches for information in its database, uses standard sets of questions and answers that help it generate the most appropriate answer, uses keywords and phrases often used in dialogues between itself and users to understand the questions, and chooses from the multitude of information, those data as appropriate as possible to the request.

#### 3.2.2. The Chatbot Implementation Process

In developing the chatbot, we had the following in mind:A.Operating system selection

Being a program, the chatbot will have to run on an operating system. The most common and used operating system on the market is Windows. Therefore, the chatbot will run on Windows.

B.Selection of programming languages

We chose to use Python because it is an easy and fast language to learn, it is free, and it contains many libraries for developing AI solutions. As an IDE, we used Spyder to write the solution in Python and run the code. We have noticed that many of the smart solutions out there today are built in Python.

C.Creating the chatbot

The chatbot is a Python-written program that uses the Seq2Seq model. Based on the Seq2Seq model and NLP techniques, it is trained to have conversations with the user.

D.Pattern matching

This technique is used by the Seq2Seq model to generate the response to the user. The data set is complex and based on this set, the Seq2Seq model will have to generate an answer.

E.Simplicity

The chatbot has a design that is easy to understand and use. The chatbot responds to users in the field in which it has been trained and only to questions asked in English. In any other case, it will generate a response indicating that it does not understand the message and asking the user to rephrase the request in English. Chatbot development is conducted in Python. The following libraries were used: Numpy [40,41], TensorFlow [42,43], Re [44], and Time [45]. Numpy has the role of processing data from arrays/matrices depending on what it wants to achieve in-app developer. TensorFlow is a software library used for machine learning and AI. TensorFlow focuses on training and inference of deep neural. Re (Regular Expression) provides regular expression matching operations, and Time provides different time-related functions.

The chatbot data set is in files with the NPY extension. These files contain arrays of data sets and information used further to train the chatbot, created with Numpy. NPY files contain data sets in an array and can be processed on any computer.

In the following there is a description of the main modules used in the implementation of the chatbot:1.clean_text()

In order for the chatbot to understand the user’s intention more easily, we have defined the clean_text function, which has the role of processing the text from the questions and answers used. Here, we have defined the following rules: both the text of the questions and the answers should contain only lowercase letters, replace abbreviations with whole words, replace special characters (-()!@#$%^&*, etc.) with string empty. Because many abbreviations are used in conversation, we have inserted the most commonly used abbreviations into the replacement patterns. Additionally, to get better chatbot responses, additional abbreviations and new rules can be introduced into this function [46,47].

Here are some examples:EU—European Union,LMK—Let me know,N/A—Not available,AKA—Also known as,Mr.—Mister,Mrs.—Misses,Ms.—Miss,N = North,E = East, andETC. = Etcetera.

These are just a few examples of abbreviations that will be replaced in the conversation for better understanding. So, if I ask: ‘Lmk if EU has improved in the last 10 years the methods for a better circular economy?’, the question it will be processed as: ‘Let me know if European Union has improve in the last 10 years the methods for a better circular economy?’.

2.tokens():

Defines the list of four tokens that the Seq2Seq model will use. The four tokens mean:-EOS represents the end of the response generated by the chatbot. Based on this added token, the Seq2Seq model will know where the response it sends to the user will end.-SOS represents the beginning of the response generated by the chatbot. Whenever the Seq2Seq model has a response to send to the user, the first thing it will do is add the string <SOS> to the response it is about to generate.-OUT has the role of replacing all the words in questions and answers that have been filtered and removed due to failure to meet the conditions regarding the established size of each statement.-PAD. To ease the process of training the chatbot, the questions and answers in the conversation will be the same size. The largest size of a question/answer is set. For all other questions/answers that do not meet these dimensions, the <PAD> string will be added until they have the established size.

Data preparation involves transforming these queries into numerical representations, such as word vectors or embedded representations. Then, if we want all sequences to be of the same length, we can add padding to the shorter questions. Let us say we set a maximum length of 10 words. In this case, questions that have this length do not require padding, but questions that do not can be filled with padding words to reach a length of 10 words.

This approach ensures that all input data are the same size and can be processed uniformly by the chatbot model. The main reasons why questions and answers need to be the same size are:-The necessity of a uniform data size

Natural language processing and machine learning models require that all input data be of the same size. Adding padding ensures that all input sequences are of the same length, regardless of the initial differences in the length of the texts. This makes it easier to process data and train models because the learning algorithm can operate more efficiently on uniformly sized data.

-The need for computational efficiency

Chatbot designs work more efficiently when the input data are of uniform size. This is because the mathematical operations applied to the data can be optimized according to their size, which leads to an acceleration of the training time and a more efficient use of computing resources.

-The need to avoid distortion of results

In the absence of padding, chatbot models might pay different attention to input sequences depending on their original length. This could lead to a distortion of the results and an incorrect weighting of the importance of different input sequences. The application of padding eliminates this problem and ensures that all input sequences are treated equally.

Additionally, there are some limitations to this approach:-Memory consumption

Adding padding increases the amount of memory required to store and process data. If you work with large data sets or extremely long texts, this can lead to significant memory usage.

-Irrelevant information

When padding to shorter texts, irrelevant or artificial information is introduced into the sequences. This can affect the performance of the chatbot model, especially when it relies on context and semantic connections between words. The more padding is added, the greater the risk of distorting the meaning of the original text.

-Additional processing time

Adding padding may result in additional processing time during training. When the data size is increased by padding, the learning algorithm will perform additional operations on these padding areas.

3.seq2seq_model():

After the data have been processed and are ready to be used, we move on to building the chatbot brain (Seq2Seq model). The feature uses the TensorFlow library for chatbot machine learning. We have defined a function for encoding the question sent by the user (encoder_rnn), a function for preprocessing the input (preprocessed_targets), and a function for decoding the response that is generated by the chatbot (decoder_rnn). All these functions are called in the seq2seq_model method [48,49].

4.training():

Initial values are set for the hyperparameters. These can then be improved by evaluating the chatbot. To train the chatbot, it is necessary to run a session that aims to create the neural network. Set the number of iterations of chatbot training. In the chatbot training algorithm, a result and a prediction error will be obtained after each iteration. The chatbot training algorithm will start with the total errors as zero, and it will increase with the error obtained after one iteration. The chatbot training algorithm sets the average error, which is calculated as the total error divided by the number of valid questions divided by the *batch_size* hyperparameter. A list is made in which all average errors obtained after each iteration will be added. If the average error obtained after the iteration is less than or equal to the minimum in the list of average errors, the chatbot is considered to have learned, speaks better, and updates the neural network with the new obtained values. Otherwise, more training is needed, and the neural network will not be updated with weaker values. If a better answer is obtained, the stop of the training algorithm will be initialized to zero; otherwise, it will be incremented by one [50,51].

#### 3.2.3. The Chatbot Training Algorithm

After the implementation of the SeqtoSeq model, the chatbot training algorithm will be developed. Figure 3 describes the steps to follow to build a chatbot that provides the correct answer. To begin with, the data are collected to train the chatbot, and various NLP techniques are applied to these data. The creation of the Seq2Seq model follows. Default values will be set for hyperparameters, values that can be changed in the future to get a chatbot that has a shorter training time and provides better answers [52].

AI is still an unused field at its true value; many options and many new, innovative solutions are experienced, transformed, analyzed, and improved. Similarly, the values can be continuously improved by testing, analyzing the test results, comparing the results obtained over different periods, and establishing strengths and weaknesses to achieve performance [53]. Once the hyperparameter values are set, the chatbot drive algorithm is implemented. Running the entire code aims to create weights based on which the chatbot can give answers. This will be a lengthy process. Other weights will be obtained at a certain time interval. If they are better than the previous ones, then they will be updated. Once you have even the first weights, you can proceed to interact with the chatbot. The first weights will not have spectacular results because the ability to understand and respond to the chatbot is limited, similar to a child. It takes more time to train the chatbot to progressively increase the chatbot’s ability to understand and respond.

After completing such a process or part of this process, we move on to interact with the chatbot and evaluate the results. New hyperparameter values can be established after the tests. It is recommended that you resume the chatbot training process to see if the new values have a positive impact on the chatbot. If the chatbot is found to give vague or incomplete responses, additional data are collected and the chatbot training process is resumed.

This chatbot is in the development phase, so the developer will determine whether the answers given by the chatbot are correct or wrong. After the training process is completed, the developer will ask the chatbot a series of questions. Based on the given answers, it will be decided if there is a need to collect additional data and establish new values of the hyperparameters.

After the answers are significantly improved, a tester will be brought in to validate the chatbot and determine which part still needs improvement and what kind of data we could add to improve the quality of the answers. Whenever new data are added or hyperparameter values are changed, the training process will have to be restarted.

Figure 4 illustrates the sequence of steps involved in obtaining a response from the chatbot. The user initiates the process by asking the chatbot a question via a messaging platform. Once the question is extracted from the platform and transmitted to the chatbot, NLP techniques come into play. These same techniques were utilized in data collection and serve to simplify the question for the chatbot’s comprehension. As seen in previous studies [54], the question is processed using NLP techniques before being presented to the Seq2Seq model, which functions as an encoder. Based on the encoder and the weights acquired during training, the chatbot generates a decoder, which constitutes the answer. This response is conveyed in the framework of the algorithm that acts as an intermediary between the messaging platform and the chatbot. Through the implementation of additional rules by the developer in this algorithm, it can be determined if the answer is correct, incomplete, or ambiguous.

If the chatbot cannot give the complete answer to the user’s question and the intervention of external services is needed to obtain a complete answer, the thought algorithm will have to call the external service after obtaining the answer of the chatbot. After calling the service, the two responses will be concatenated to get the last answer provided to the user. If the chatbot provides the default answer, human intervention is needed. Therefore, it will be notified to a human to provide an answer to the user, and the chatbot will provide a default response to the user.

We have described above the steps we intend to follow to build the chatbot. Our interests consist of creating a mobile platform/application in the field of the circular economy and also a chatbot that will be able to dialogue with the user and provide useful information. Our research is inspired by the most recent developments in neural translation, using the attention mechanism to learn long sequences and improve response performance. The Seq2Seq model generates a word with the help of information collected in distinct parts of the sentence instead of searching for the entire sentence.

## 4. Conclusions

The principal findings of this article are as follows:An analysis and presentation of five chatbots, which have been developed as solutions within the CE;A comparison of the functionalities of these five chatbots that have been previously scrutinized;A detailed exposition of the development process for a chatbot that caters to the CE;A conclusive summary that highlights the feasibility of employing AI-based solutions to benefit CE.

We contend that our study constitutes a compelling argument for the integration of AI in the circular economy, particularly in light of the contemporary proliferation of chatbots such as ChatGPT [55], Chatsonic [56] and Youchat [57]. The remarkable technological strides being made in the domain of AI show a concerted effort to substitute the human factor in tasks that were once deemed exclusive to humans, with notable success.

In addition, in the field of CE, such solutions based on artificial intelligence can be implemented [58], with the following key points as the main objectives:-Reduction of waste and pollution: CE tries to reduce the amount of waste generated by designing products in a way that they can be reused, repaired, or recycled. This helps reduce pollution, improves air and water quality, and reduces environmental impact.-Saving resources: The circular economy aims to use resources more efficiently, by extending the life of products and using materials and resources sustainably. This can reduce costs and improve the efficiency and sustainability of the economy.-Increasing efficiency: The circular economy encourages the use of resources in a smarter and more productive way to maximize economic and social benefits and minimize environmental impact. This can help increase the productivity and competitiveness of companies and contribute to the creation of new jobs.-Promoting innovation: The circular economy encourages innovation in the design, production and use of products and materials to improve the efficiency and sustainability of the economy [59]. This can generate new business opportunities and help develop innovative solutions to social and environmental problems.

Based on the content, we consider that the present work offers a starting point for further studies of applications of artificial intelligence in the field of CE. In addition, in this article, we outline a way to design and create a chatbot plan that can be used as an AI application in CE. This paper presented and analyzed five chatbot platforms and their implementation aimed at attracting the attention of the population to a cleaner and less polluted environment.

The first example is a platform designed by TheCircularLab in the city of Logroño (Spain). TheCircularLab uses the latest features to take over and provide information about the various resources that can be recycled, about the places where recycling is conducted, and also general information about what the circular economy means and its need. The uniqueness of the application is given by the ability to recognize the image, which allows the application to be more accurate and use visual stimuli to perform the task that it has to solve. The second example is a platform developed in Yokohama, Japan. The name of the chatbot is Iio and it is made by NTT Docomo specialists. The key role of the chatbot is to help classify waste and materials, as to well as direct the population to proper recycling. The third example is a platform developed by the French company Trizzy, which is a revolutionary product in this field. Trizzy is the first assistant to incorporate the concepts of organizations and communities with three principal areas of action: social ties (the connection between people and the local ecosystem), the circular economy, and waste (as well as strengthening a change in behavior in this regard). The fourth example is BioHiTech Alto, one of the first chatbots dedicated to waste reduction. The last chatbot example that we have analyzed in this paper is Eco-bot, an application that acts similar to a virtual energy-saving assistant.

However, the implementation of such a multifaceted solution presents a challenging endeavor. In this regard, we have outlined a series of procedures, starting from the identification of the target deployment area and proceeding with an extensive and continuous effort toward gathering, examining, interpreting, categorizing, and filtering information by specialized analysts. Moreover, we have introduced several methodologies employed in the design of the platform, which include the utilization of the Seq2Seq model, as well as chatbot training algorithms that leverage natural language processing techniques for data processing and conversion.

In the upcoming period, our plan is to conceive and construct a comprehensive solution in the form of a chatbot-like platform that integrates all the aforementioned functionalities. To achieve this, we must acquire data that will power the application and enable the chatbot to engage in a conversation with a human interlocutor, while delivering intelligible responses. Furthermore, the chatbot must be equipped with relevant information on the subject of the circular economy and its potential to contribute to a cleaner environment without pollution. This will streamline the provision of information in the targeted discussion domain.

## Figures and Tables

**Figure 1 sensors-23-04990-f001:**
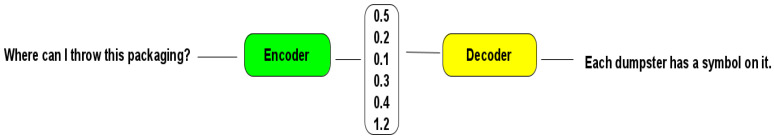
The SeqToSeq model.

**Figure 2 sensors-23-04990-f002:**
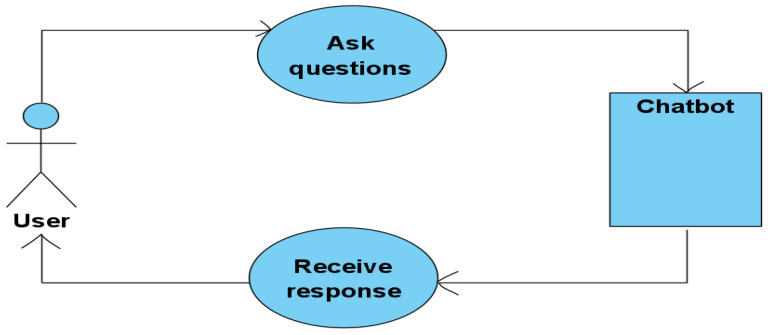
Use case diagram of chatbot design.

**Figure 3 sensors-23-04990-f003:**
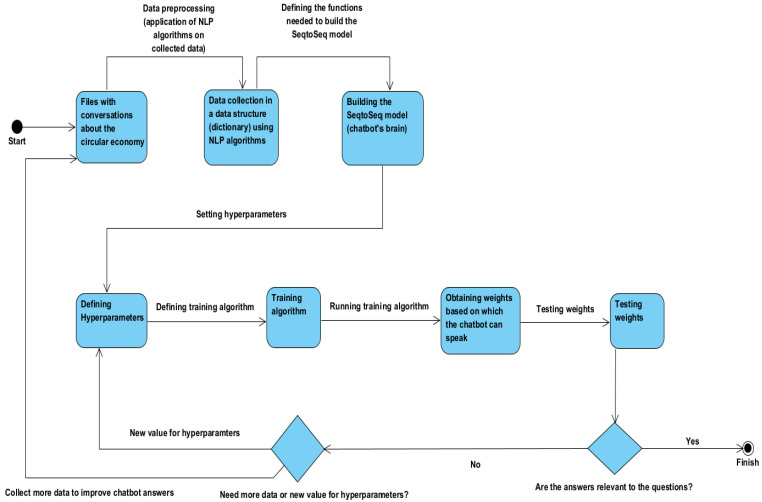
Chatbot training process.

**Figure 4 sensors-23-04990-f004:**
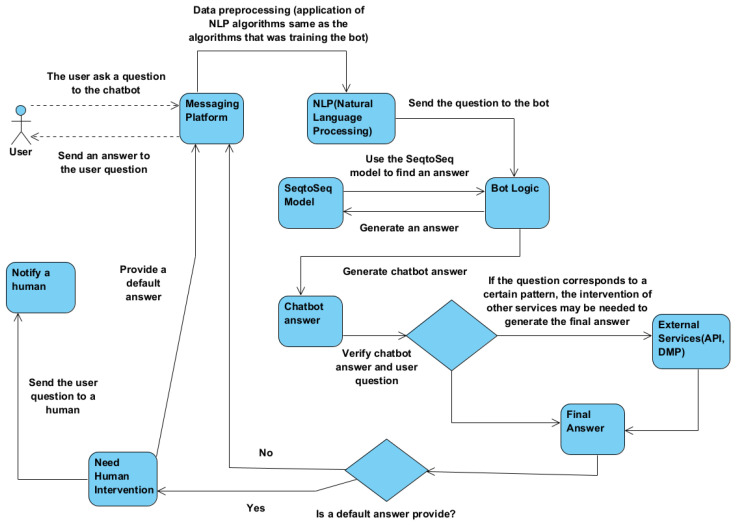
Chatbot generate answer process.

**Table 1 sensors-23-04990-t001:** Chatbots functionalities.

Chatbot	NLP	Security *	Dialogue *	UI *	Data Acc. *
A.I.R-e	✔	7	6	8	8
Iio	✔	8	7	7	6
Trizzy	✔	7	6	8	7
BioHiTech	✔	8	5	8	6
Eco-Bot	✔	7	5	8	6

* Out of ten.

## Data Availability

Data is contained within the article.

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
