# Peer review of "Use and Design of Chatbots for the Circular Economy [Author-notes fn1-sensors-23-04990]"

_sensors, 2023, doi:10.3390/s23114990_

Round 1

Reviewer 1 Report

This manuscript presents an analysis of how Information Technology and Communications (ITC) can help consumers on the issue of transition to the Circular Economy (CE) in the field of recycling materials and equipment. The authors presented and analyzed five chatbot solutions used in the field of CE, and then designed the procedures for data collection, training, development, and testing of a chatbot using various natural language processing (NLP) and deep processing (DP) techniques.

I have the following comments.

1. Line 517: “Table 1. Chatbots functionalities.”

Please give details on how the points (out of 10) were calculated. This is not explained in detail in the text.

2. Line 604: “Because many abbreviations are used in conversation, we have inserted the most commonly used abbreviations into the replacement patterns. Additionally, to get better chatbot responses, additional abbreviations and new rules can be introduced into this function.”

Please add some examples to clarify this point. It is not clear how abbreviations are handled.

3. Line 619: “PAD. To ease the process of training the chatbot, the questions and answers in the conversation will be the same size.”

This restriction limits the chatbot’s ability by restricting the length of its generated answers. Could you please prove why this restriction is necessary? Also, please comment on the limitations introduced by this precondition on the sizes of questions and answers.

4. Line 672: Figure 3: At the end of Figure 3, if the answers are “wrong,” the figure shows the step of  “Assigning other values to hyperparameters and resuming the training process.” Who judges or decides that the answers are wrong? This step is unclear and needs more clarification in the text with an illustrative example. Also, the corresponding figures should be updated to illustrate the details of (i.e., steps of) dealing with wrong answers.

5. Line 689: “If the chatbot's response is unclear, the algorithm may find that human intervention is needed by notifying the development team and providing the user with an error response (a sign that it cannot understand the user's message).

Figure 4 shows that “human intervention” is part of “External Services;” however, in Line 689, human intervention is described as taking place after providing the answer to the user. Are there two (2) stages (or steps) of human intervention? I mean, one before providing the answer and one after providing the answer. This point is not clear. Please explain this in the text and in Figure 4.

6. Line 708: “This section is not mandatory but can be added to the manuscript if the discussion is unusually long or complex.” Is this part of the manuscript (it seems to be a side note)?

7. The main contribution of this manuscript is not clear. Please add a numbered list that outlines the main contributions. (also compare the contributions to similar published papers).

Extensive editing of English language is required (some parts are difficult to understand).

Author Response

Hello, first of all thank you for your comments and suggestions; here are our answers:

1. Line 517: “Table 1. Chatbots functionalities.”

Please give details on how the points (out of 10) were calculated. This is not explained in detail in the text.

Ans: Section 3.1.6 was renamed to "Chatbots comparison and analysis" (line 512)
We added here some basic conclusions about those five chatbot solutions that were examined. Also, we added an explanation about how the points were calculated. (lines 513-547)

2. Line 604: “Because many abbreviations are used in conversation, we have inserted the most commonly used abbreviations into the replacement patterns. Additionally, to get better chatbot responses, additional abbreviations and new rules can be introduced into this function.”

Please add some examples to clarify this point. It is not clear how abbreviations are handled.

Ans: At lines 637- 652 we added some examples for a better understanding

3. Line 619: “PAD. To ease the process of training the chatbot, the questions and answers in the conversation will be the same size.”

This restriction limits the chatbot’s ability by restricting the length of its generated answers. Could you please prove why this restriction is necessary? Also, please comment on the limitations introduced by this precondition on the sizes of questions and answers.

Ans: At lines 675- 708 we provide the reasons why we need this restriction. Also we provide some limitations of this restriction. 

4. Line 672: Figure 3: At the end of Figure 3, if the answers are “wrong,” the figure shows the step of  “Assigning other values to hyperparameters and resuming the training process.” Who judges or decides that the answers are wrong? This step is unclear and needs more clarification in the text with an illustrative example. Also, the corresponding figures should be updated to illustrate the details of (i.e., steps of) dealing with wrong answers.

Ans: At lines 760-769 there is explanation on who judge the answers of the chatbot. Also, figure 3 was updated. 

5. Line 689: “If the chatbot's response is unclear, the algorithm may find that human intervention is needed by notifying the development team and providing the user with an error response (a sign that it cannot understand the user's message).

Figure 4 shows that “human intervention” is part of “External Services;” however, in Line 689, human intervention is described as taking place after providing the answer to the user. Are there two (2) stages (or steps) of human intervention? I mean, one before providing the answer and one after providing the answer. This point is not clear. Please explain this in the text and in Figure 4.

Ans: Lines 773- 784 where modified, also figure 4 was updated 

6. Line 708: “This section is not mandatory but can be added to the manuscript if the discussion is unusually long or complex.” Is this part of the manuscript (it seems to be a side note)?

Ans: This part was removed

7. The main contribution of this manuscript is not clear. Please add a numbered list that outlines the main contributions. (also compare the contributions to similar published papers).

Ans. In the last part of the article, entitled Conclusions, we added a numbered list containing the main contributions (lines 800-808):

"The principal findings of this article are as follows:

  1. An analysis and presentation of five chatbots, which have been developed as solutions within the CE;
  2. A comparison of the functionalities of these five chatbots that have been previously scrutinized;
  3. A detailed exposition of the development process for a chatbot that caters to the CE;
  4. A conclusive summary that highlights the feasibility of employing AI-based solutions to benefit the CE."

We have also improved English readability throughout the text.

Obs. Please excuse us, but the line numbers may not match exactly in all cases, considering the many changes we have made. Thank you!

Reviewer 2 Report

Very interesting topic indeed, please respond to the following:

Do you plan a code of rules in order to protect humans by erroneous IA decisions?

How do you test IA reasoning capabilities against humans?

Are you sure a machine understands the needs of a sustainable economy better than a human?

What security protocols have you in mind the bot not to be hacked by terrorists?

How do you plan to create a scenarios database since our recording of climate changes is so... young?

Author Response

Hello, thank you for your comments; here are the answers to your questions:

- Do you plan a code of rules in order to protect humans by erroneous IA decisions?

We didn't plan to include a code of rules in our present article, but in the future development process of a chatbot we would like to include the following guideline principles: 

1. Ethical Guidelines and Principles
Various organizations and institutions have formulated ethical guidelines and principles for AI development.
2. Standardization and Certification
Efforts are being made to develop standards and certification processes for AI systems. These initiatives aim to ensure that AI systems meet specific requirements related to fairness, safety, transparency, and accountability. 
3.Public Awareness and Education
Promoting public awareness and understanding of AI technologies and their potential impacts is crucial. By fostering education and public engagement, individuals can better comprehend AI systems, their limitations, and the ethical considerations surrounding their use.

- How do you test IA reasoning capabilities against humans?
In the future development of our chatbot we may include some of the following methods:
1. Using benchmark Datasets
Researchers often create benchmark datasets that contain a wide range of questions or tasks that require reasoning. These datasets are designed to evaluate AI systems' ability to reason and provide accurate answers or solutions. Human performance on these benchmarks serves as a baseline for comparing AI system performance.
2. Using human Evaluators
Human evaluators can be involved in directly assessing AI reasoning capabilities. They may be asked to review and rate the quality of AI-generated responses or solutions. These evaluations can be subjective or objective, depending on the specific task. Human evaluators can provide valuable insights into the reasoning abilities of AI systems by comparing their performance to human performance.
3. Adversarial Testing
Adversarial testing involves creating challenging scenarios or inputs specifically designed to test the reasoning capabilities of AI systems. These scenarios often aim to expose weaknesses or biases in the AI's reasoning process. Adversarial testing can be performed by humans who actively try to deceive or confuse the AI system, or by using specially crafted inputs that require sophisticated reasoning.
4. Turing Test
The Turing Test, proposed by Alan Turing, is a classic test of AI reasoning capabilities. It involves a human evaluator engaging in a conversation with an AI system and another human simultaneously. If the evaluator cannot reliably distinguish between the AI system's responses and the human's responses, the AI system is considered to exhibit reasoning capabilities that are indistinguishable from those of a human.

- Are you sure a machine understands the needs of a sustainable economy better than a human?

This is an interesting and complex question! The answer involves multiple factors:
1. Data Analysis
Machines can process vast amounts of data and perform complex analyses at a speed and scale that surpass human capabilities. This can allow them to identify patterns, trends, and correlations that humans may miss, leading to more informed insights into sustainable economic practices.
2. Objective Decision-Making
Machines can make decisions based on predefined rules, algorithms, and data, without being influenced by emotions, biases, or subjective factors. This objectivity can potentially contribute to more consistent and unbiased decision-making in the context of a sustainable economy.
3. Human Context and Values
On the other hand, humans possess contextual understanding, critical thinking, and the ability to consider ethical, social, and cultural factors. Humans can integrate various perspectives, assess long-term implications, and make value-based judgments that machines may struggle with.
4. Interdisciplinary Approach
Addressing the needs of a sustainable economy requires a multidisciplinary approach that encompasses economic, social, environmental, and ethical dimensions. Combining human expertise from diverse fields can facilitate a holistic understanding and decision-making process.
5. Ethical Considerations
Decisions related to a sustainable economy involve ethical judgments and trade-offs that extend beyond pure data analysis. Determining the value of different aspects of sustainability, such as environmental preservation, social equity, or economic growth, often requires subjective and value-based assessments that go beyond the scope of machines.

- What security protocols have you in mind the bot not to be hacked by terrorists?

Speaking about security, this is accomplished by the default operating system and browser's security implementation.

- How do you plan to create a scenarios database since our recording of climate changes is so... young?

Even though scientists' observations of climate change are relatively recent, different scenarios have been developed based on historical data, records and paleoclimate models. These researches can help us create a database of scenarios regarding climate change and future forecasts.

Reviewer 3 Report

The article deals with the application of chatbots for private users in the context of the circular economy, in particular in the context of handling of waste. The authors used case studies, Participant observation; Focus on the group; Thematic study of documents as analytical approaches.

In the paper, five examples of chatbots for the circular economy are presented and explained as to their conception and their thematic focus.

The paper is very illustrative as introduction to the practical use of chatbots and their technical conception. Although the authors claim to put a focus on participant observation, the real use for private users, the possible technical problems, the acceptance of the service remains unclear.

To conclude, the paper is very interesting, but needs an extension as to the practical application of chatbots.

Author Response

Thank you for your comments!

Regarding your last comment that the paper "needs an extension as to the practical application of chatbots.", we plan to develop such an application in the near future.

We have also specified more clearly in the conclusions section the main findings of our paper:

"The principal findings of this article are as follows:

  1. An analysis and presentation of five chatbots, which have been developed as solutions within the CE;
  2. A comparison of the functionalities of these five chatbots that have been previously scrutinized;
  3. A detailed exposition of the development process for a chatbot that caters to the CE;
  4. A conclusive summary that highlights the feasibility of employing AI-based solutions to benefit the CE."

Round 2

Reviewer 1 Report

The authors have adequately addressed the comments. 

Thank you.

Minor editing of English language required.

Reviewer 2 Report

Thank you for your kind answers, great job!

Reviewer 3 Report

The paper has improved considerably. It is an interesting introduction in the use of chatbots. Now the paper can be published without further changes.The paper has improved considerably. It is an interesting introduction in the use of chatbots. Now the paper can be published without further changes.